# Vibration Characteristics and Experimental Research of Combined Beam Tri-Stable Piezoelectric Energy Harvester

**DOI:** 10.3390/mi13091465

**Published:** 2022-09-04

**Authors:** Xuhui Zhang, Hengtao Xu, Xiaoyu Chen, Fulin Zhu, Yan Guo, Hao Tian

**Affiliations:** 1College of Mechanical Engineering, Xi’an University of Science and Technology, Xi’an 710054, China; 2Shaanxi Key Laboratory of Mine Electromechanical Equipment Intelligent Monitoring, Xi’an University of Science and Technology, Xi’an 710054, China

**Keywords:** tri-stable, combined beam, nonlinearity, static bifurcation, dynamic response

## Abstract

Vibration energy harvesting technology is expected to solve the power supply and endurance problems of wireless sensor systems, realize the self-power supply of wireless sensor systems in coal mines, and promote the intelligent development of coal mine equipment. A combined beam tri-stable piezoelectric energy harvester (CTPEH) is designed by introducing magnetic force into the combined beam structure. In order to explore the vibration characteristics of CTPEH, a nonlinear magnetic model is obtained based on the magnetic dipole theory, and the dynamic equation of the system is established using the Lagrange theorem and Rayleigh–Ritz theory. The influence of the different magnet distances and excitation conditions on the static bifurcation characteristics and dynamic response characteristics of the system are analyzed by numerical simulation, and the simulation results are validated by the experiments. The research results show that the motion state of the CTPEH system has four transition forms from mono-stable to tri-stable with the change in magnet distance. The tri-stable system has three potential energy curves with different characteristic shapes. The appropriate starting excitation position and excitation frequency can make it easier for the system to realize a large-amplitude response state, thereby improving the output performance of the system. This research provides new ideas and methods for optimizing the performance of the combined beam piezoelectric energy harvester.

## 1. Introduction

In recent years, with the increasing maturity of wireless sensor network and micro-electro-mechanical system (MEMS) technology, wireless sensor systems have been widely applied in many fields such as industry [1], transportation [2], and military [3]. At present, the power supply method of the wireless sensor system is mainly traditional batteries. Due to volume and capacity constraints, the electric energy provided by traditional batteries is very limited, as well as represents an environmental pollution risk. Therefore, as a key technology to replace traditional batteries, vibration energy harvesting technology enables the realization of self-powered wireless sensor systems.

The current vibration energy harvesting methods mainly include electromagnetic [4], electrostatic [5], magnetoelectric [6], and piezoelectric methods [7]. The piezoelectric type is more suitable for various applications than other energy harvesting methods due to its easy integration and high efficiency [8]. Piezoelectric energy harvesters are mainly cantilever beam structures, and researchers have extensively analyzed and researched this structure [9,10]. However, the traditional cantilever beam energy harvester can only harvest energy when the vibration frequency of the external environment is the same as its resonant frequency, which leads to its low energy harvesting efficiency. Therefore, researchers have proposed many methods to broaden the frequency band so that the energy harvester can adapt to the broadband characteristics of the external environment [11]. Frequency extension methods are mainly divided into linear and nonlinear [12]. Linear extension methods mainly include array structure [13], multi-degree-of-freedom structure [14], special-shaped structure [15], etc. Although these methods can realize frequency extension, they have disadvantages such as complex systems and large structure space occupation. The methods of nonlinear frequency extension mainly include adding a stopper [16], introducing a spring [17], magnetic coupling [18], etc. Related researchers have researched the application of these methods. Halim [19] designed a piezoelectric energy harvester with a base stopper. The research results showed that the working frequency band of the energy harvester is greatly improved compared with that without a stopper. Aladwani [20] proposed a cantilever beam energy harvester with the base connected to a spring. The results indicate that the system realizes the broadening of the energy harvesting frequency band using the amplification effect of the spring.

Compared with the two above nonlinear frequency extension methods, magnetic coupling has the advantages of system stability and a simple structure, so researchers have conducted in-depth discussions and research on the magnetic coupling piezoelectric energy harvester [21,22]. Shah [23] designed a bi-stable piezoelectric energy harvester by introducing two mutually repelling magnets. The simulation and experimental results show that the output performance of the bi-stable structure is significantly improved compared with the traditional linear structure. Li [24] proposed a magnetically coupled tri-stable piezoelectric energy harvester and established a dynamics model of the system. It was demonstrated through numerical simulations and experiments that the tri-stable structure can produce higher output voltage and power under the same excitation conditions as the bi-stable energy harvesting structure. Ma [25] constructed an asymmetric tri-stable piezoelectric energy harvester by taking an asymmetric placement of two magnets on the base. The analysis showed that the asymmetric tri-stable structure has complex response characteristics, and it is easier to achieve large-amplitude movements than the symmetric structure, thus enhancing the output efficiency of the system. Zhou [26] improved the tri-stable piezoelectric energy harvester by designing the fixed magnet to be rotatable and showed that the fixed magnet, when rotated to a suitable angle, can significantly enhance the operating band and harvesting efficiency of the harvester. Ju [27] added a magnet based on the tri-stable structure to form a quad-stable piezoelectric energy harvester. According to simulation and experimental studies, the quad-stable piezoelectric energy harvester has a shallower potential well, for which it is easier to realize interwell motion, and the system’s frequency bandwidth and output performance are further improved. Zhou [28] designed a tri-stable piezoelectric cantilever energy harvester and studied the system response characteristics at different initial positions through frequency sweep experiments. The study found that the initial position has a great influence on the effective frequency range of the system. Wang [29] proposed a bi-stable piezoelectric energy harvester with an asymmetric potential well and analyzed the effect of initial displacement on the system’s performance. The results indicate that the system is more likely to exhibit a high-energy output state when the initial displacement is at a shallower potential well.

The vibration in the external environment is often in multiple directions, while the energy harvesting direction of the straight beam structure is single, which cannot effectively harvest the multidirectional vibration energy in the environment. Therefore, researchers have gradually studied the curved beam energy harvesting structure [30,31]. Chen [32] designed an arched beam-type energy harvesting structure. Compared with the straight beam structure, the arched beam has a larger average strain, which can improve the output performance of the energy harvester. Jung [33] proposed a curved piezoelectric energy harvesting structure. The experimental study showed that the curved structure not only expands the vibration frequency range of the system, but also improves the power generation of the system. Zhao [34] designed an arc-shaped elastic energy harvesting structure. Through finite element analysis and experimental tests, it was shown that the arc-shaped structure can effectively harvest energy in multiple directions in the environment.

Regarding the wide frequency band and multi-direction vibration characteristics of electrical and mechanical mine equipment, this research group proposed a combined beam energy harvesting structure in the early stage. Based on the combined beam structure and magnetic coupling, a combined beam tri-stable piezoelectric energy harvester (CTPEH) was designed in this paper. The work of this paper is organized as follows. In Section 2, the overall structure of CTPEH is first introduced, and then the nonlinear magnetic model and restoring force model of the system are provided; finally, the motion model of the system is established based on the Lagrange principle. In Section 3, the influence of the magnet distance on the static bifurcation and potential energy characteristics of the system is analyzed, and then the influence of the starting position and excitation frequency on the dynamic response of the system is numerically simulated. In Section 4, the experimental platform and prototype of the system are first introduced, and then experiments are conducted to verify the simulation results. The main conclusions of this paper are given in the last section.

## 2. Structure and Theoretical Model of CTPEH

### 2.1. Structure of CTPEH

Figure 1 shows the schematic diagram of the structure of CTPEH, which mainly consists of a base, a combined beam, piezoelectric material, and magnets. The combined beam is composed of a linear straight beam and an arched curved beam, one end of which is fixed on the inner wall on the left side of the base, whilst the other end is free. The piezoelectric material is a polyvinylidene fluoride (PVDF) piezoelectric film, which is attached to the upper surface of the combined beam. Its positive and negative electrodes are connected to both ends of the load resistor R. Magnet A is fixedly connected to the free end of the combined beam, whilst the magnets B and C are symmetrically fixed to the inner wall of the right side of the base about the *x* axis, and the magnetic forces between magnet A and magnets B and C repel each other. As shown in the figure, the transverse length of the combined beam is L, the surface distance between magnet A and magnets B and C along the *x* axis is $d_{x}$, and the center distance between magnet B and magnet C along the *y* axis is $d_{y}$. When the base is excited along the *y* axis direction, the combined beam will bend, and the piezoelectric film on the surface will also deform to generate electricity.

### 2.2. Theoretical Model of CTPEH

#### 2.2.1. Nonlinear Magnetic Force Model

To analyze the dynamic response characteristics of CTPEH, the nonlinear magnetic force between magnet A and magnets B and C needs to be calculated accurately. In this paper, the magnetic force model of the system is built based on the magnetic dipole theory, and the position relationship between magnets A, B, and C is schematically shown in Figure 2.

The magnetization strength of magnets A, B, and C are $M_{A}$, $M_{B}$, and $M_{C}$, respectively. $S_{A}$, $S_{B}$, and $S_{C}$ are the surface areas of the three magnets, respectively. The length of the magnet along the *x* axis is e. According to the magnetic dipole method, the magnetic repulsion force of magnet A on magnet B is [35]
(1)$$F_{AB}=\frac{K_{AB}}{4\pi }(-\frac{\lambda_{1}}{{\left({\left(d_{x}+e\right)}^{2}+\lambda_{1}^{2}\right)}^{\frac{3}{2}}}+\frac{\lambda_{1}}{{\left({\left(d_{x}+2e\right)}^{2}+\lambda_{1}^{2}\right)}^{\frac{3}{2}}}+\frac{\lambda_{2}}{{\left(d_{x}^{2}+\lambda_{2}^{2}\right)}^{\frac{3}{2}}}\;-\frac{\lambda_{2}}{{\left({\left(d_{x}+e\right)}^{2}+\lambda_{2}^{2}\right)}^{\frac{3}{2}}})$$
where $K_{AB}=\mu_{0}M_{A}S_{A}M_{B}S_{B}$, $\lambda_{1}=u\left(L,t\right)-d_{y}/2$, $\lambda_{2}=u\left(L,t\right)+\theta e-d_{y}/2$. $\mu_{0}$ is the vacuum magnetic permeability; u(L,t) is the vibration displacement of magnet A along the *y* axis direction; and θ is the deflection angle of magnet A relative to the *x* axis.

Similarly, the magnetic repulsion force of magnet A by magnet C can be expressed as
(2)$$F_{AC}=\frac{K_{AC}}{4\pi }(-\frac{\lambda_{3}}{{\left({\left(d_{x}+e\right)}^{2}+\lambda_{3}^{2}\right)}^{\frac{3}{2}}}+\frac{\lambda_{3}}{{\left({\left(d_{x}+2e\right)}^{2}+\lambda_{3}^{2}\right)}^{\frac{3}{2}}}+\frac{\lambda_{4}}{{\left(d_{x}^{2}+\lambda_{4}^{2}\right)}^{\frac{3}{2}}}\;-\frac{\lambda_{4}}{{\left({\left(d_{x}+e\right)}^{2}+\lambda_{4}^{2}\right)}^{\frac{3}{2}}})$$
where $K_{AC}=\mu_{0}M_{A}S_{A}M_{C}S_{C}$, $\lambda_{3}=u\left(L,t\right)+d_{y}/2$, $\lambda_{4}=u\left(L,t\right)+\theta e+d_{y}/2$.

Then, the total magnetic repulsion force of magnet A on magnets B and C is
(3)$$F_{M}=F_{AB}+F_{AC}$$

Therefore, the total potential energy generated between the magnets can be derived as:(4)$$U_{M}=\int^{\;}F_{M}du$$

#### 2.2.2. Nonlinear Restoring Force Model

The restoring force of the combined beam in this paper is relatively complicated due to the existence of curved beam parts. The traditional straight beam is generally considered a linear restoring force, while the restoring force of the combined beam is nonlinear. This paper used a dynamometer (YLK-10, ELECALL, Yueqing, China) to survey the restoring force of the combined beam several times, and then took the mean value of the measured data, and finally used MATLAB (version 9.1, accessed on 15 June 2022) to perform a nonlinear curve fitting. The polynomial expression of the restoring force–displacement is obtained as follows:(5)$$F_{R}=c_{1}u^{3}\left(L,t\right)+c_{2}u\left(L,t\right)$$
where $c_{1}$ and $c_{2}$ are the coefficients of the cubic term and the primary term of the expression, respectively.

The measured data and curve fitting results of the nonlinear restoring force of the combined beam are shown in Figure 3, which obtains the coefficient $c_{1}=55,314.7\;N/m^{3}$, $c_{2}=20.45\;N/m$. It can be seen from Figure 3 that the nonlinear curve fitting is relatively good.

#### 2.2.3. Dynamic Model of CTPEH

In this paper, the dynamic model of CTPEH is established according to the Euler–Bernoulli theory and Lagrange theorem. This paper mainly studies the vibration characteristics of the combined beam in the y direction. In this direction, the tensile deformation of the curved beam part cannot be considered, and the deformation of the combined beam is similar to that of the straight beam. Therefore, the combined beam is regarded as a straight beam in the dynamic model of this paper. The Lagrange equation of the system can be expressed as
(6)$$L\left(x,t\right)=T_{L}+T_{M}+W_{P}-U_{M}-U_{R}$$
where $T_{L}$ is the kinetic energy of the piezoelectric material layer and the substrate layer; $T_{M}$ is the kinetic energy of free-end magnet A; $W_{P}$ is the electrical energy generated by the piezoelectric material; $U_{M}$ is the magnetic potential energy generated between the magnets; $U_{R}$ is the strain potential energy of the piezoelectric material layer and the substrate layer. Their specific expressions are as follows:(7)$$T_{L}=\frac{1}{2}\left(\rho_{P}A_{P}+\rho_{S}A_{S}\right)\int_{0}^{L}{\left[\frac{\partial u\left(x,t\right)}{\partial x}+\dot{z}\left(t\right)\right]}^{2}dx$$
(8)$$T_{M}=\frac{1}{2}M_{t}{\left\{{\left[\frac{\partial u\left(x,t\right)}{\partial t}\right]}_{x=L}+\dot{z}\left(t\right)\right\}}^{2}+\frac{1}{2}I_{t}{\left[\frac{\partial^{2}u\left(x,t\right)}{\partial t\partial x}\right]}_{x=L}^{2}$$
(9)$$W_{P}=\frac{1}{4}e_{31}b\left(h_{S}+h_{P}\right)v\left(t\right){\left[\frac{\partial u\left(x,t\right)}{\partial x}\right]}_{x=L}+\frac{1}{2}C_{P}v^{2}\left(t\right)$$
(10)$$U_{R}=\int^{\;}F_{R}du$$
where $\rho_{P}$ and $A_{P}$ are the density and cross-sectional area of the piezoelectric layer, respectively; $\rho_{S}$ and $A_{S}$ are the density and cross-sectional area of the substrate layer; $\dot{z}\left(t\right)$ is the excitation speed of the base; $M_{t}$ and $I_{t}$ are the mass and rotational inertia of magnet A, respectively; $e_{31}$ is the electromechanical coupling constant of the piezoelectric material; $C_{P}$ is the equivalent capacitance of the piezoelectric material.

In this paper, we mainly focus on the energy harvesting problem in a low-frequency excitation environment, so only the first-order vibration mode of the combined beam is considered. The Rayleigh–Ritz theory can be used to express the vibration displacement u(x,t) as
(11)u(x,t)=ϕ(x)η(t)
where ϕ(x) is the first-order mode shape of the beam, η(t) is the first generalized modal coordinate. The combined beam structure in this paper is a cantilever beam, so the allowable function of the cantilever beam is used to approximate the modal mode shape of the combined beam as follows [36]:(12)$$\phi \left(x\right)=1-\cos(\frac{\pi x}{2L})$$

According to Lagrange’s theorem and Kirchhoff’s current law, combined with the nonlinear magnetic model and restoring force model of the system, the dynamic equation of the system can be obtained as
(13)$$M\ddot{\eta }\left(t\right)+C\dot{\eta }\left(t\right)+F_{M}+F_{R}-\vartheta v\left(t\right)=-\psi \ddot{z}\left(t\right)$$
(14)$$\vartheta \dot{\eta }\left(t\right)+C_{P}\dot{v}\left(t\right)+v\left(t\right)/R=0$$
where M is the modal mass of the system, ϑ is the electromechanical coupling coefficient term, ψ is the excitation coefficient term, and $\ddot{z}\left(t\right)$ is the acceleration of the external excitation. Their specific expressions are given below:(15)$$M=\left(\rho_{P}A_{P}+\rho_{S}A_{S}\right)\int_{0}^{L}\phi^{2}\left(x\right)dx+M_{t}\phi^{2}\left(L\right)+I_{t}{\left(\phi^{\prime }\left(L\right)\right)}^{2}$$
(16)$$\vartheta =\frac{1}{2}e_{31}b\left(h_{P}+h_{S}\right)\phi^{\prime }\left(L\right)$$
(17)$$\psi =\left(\rho_{P}A_{P}+\rho_{S}A_{S}\right)\int_{0}^{L}\phi \left(x\right)dx+M_{t}\phi \left(L\right)$$
(18)$$\ddot{z}\left(t\right)=Asin\left(2\pi Ft\right)$$

In Equation (18), A is the external excitation amplitude and F is the external excitation frequency.

## 3. Numerical Simulation of CTPEH

### 3.1. Analysis of Static Bifurcation Characteristics

The structure and material parameters of the substrate layer, piezoelectric layer, and magnet in the CTPEH system are shown in Table 1.

Let the parameters of the time term in Equation (13) be zero, that is, $\ddot{\eta }\left(t\right)=\dot{\eta }\left(t\right)=\dot{v}\left(t\right)=v\left(t\right)=\ddot{z}\left(t\right)=0$, and the expression for the static solution of the system can be obtained as
(19)$$F_{M}+F_{R}=0$$

According to Equation (19), the static solution of the system in the space of ($d_{x}$, $d_{y}$, u) is schematically shown in Figure 4. It can be seen that the system exhibits complex bifurcation characteristics. With the change of the magnet distance $d_{x}$ and $d_{y}$, the system has three situations of one, three, and five equilibrium positions.

Projecting Figure 4 on the ($d_{x}$, $d_{y}$) plane obtains the bifurcation set of the system’s equilibrium position, as shown in Figure 5. Divide Figure 5 into three regions: A, B, and C. When $d_{x}$ and $d_{y}$ are in region A, the system has only one equilibrium position and exhibits a mono-stable state. In region B, the system has three equilibrium positions, which is a bi-stable system at this time. When located in region C, there are five equilibrium positions for the system, and the system exhibits tri-stable characteristics. The curve ab is the dividing line between region A and region B. When $d_{x}$ and $d_{y}$ are on the curve ab, the system is in the critical state between being mono-stable and bi-stable. Similarly, the curve ac is the dividing line between region A and region C. When $d_{x}$ and $d_{y}$ are on the curve ac, the system is in the critical state between mono-stable and tri-stable. The curve ad is the dividing line between region B and region C. When $d_{x}$ and $d_{y}$ are on the curve ad, the system is in the critical state between being bi-stable and tri-stable.

The magnet distance has a significant effect on the stable motion of the system, which is caused by the phenomenon of static bifurcation. In the following, the static bifurcation characteristics of the system are analyzed in detail by numerical simulation.

The effect on the static bifurcation characteristics of the system in the ($d_{x}$, u) plane is first analyzed. Figure 6 shows the static bifurcation diagram of the system when $d_{y}=0,\;10,\;12,\;16\;\mathrm{mm}$. The red solid line in the figure indicates the stable equilibrium position, and the blue dashed line indicates the unstable equilibrium position. As shown in Figure 6a, when $d_{y}=0\;\mathrm{mm}$, the bifurcation phenomenon of the system at this time is similar to the bi-stable structure, and there is only one bifurcation point P_1_ in the system. When the value of $d_{x}$ is taken on the right side of the bifurcation point P_1_, there is only one stable equilibrium position in the system, and the system behaves as a mono-stable. When the value is taken on the left side of point P_1_, the system has three equilibrium positions, two of which are stable and one which is unstable. At this time, the system is a bi-stable system. As $d_{y}$ increases to 10 mm, it can be seen from Figure 6b that the bifurcation point P_1_ moves to the left, and a new bifurcation point P_2_ appears on the left side of P_1_. The bifurcation points P_1_ and P_2_ divide $d_{x}$ into three intervals. When $d_{x}>d_{xP1}$, the system has only one stable equilibrium position, and the system presents mono-stable characteristics. When $d_{xP1}>d_{x}>d_{xP2}$, the system has two stable and one unstable equilibrium position and is in a bi-stable state. When $d_{x}<d_{xP2}$, the system has three stable and two unstable equilibrium positions and exhibits tri-stable characteristics.

As shown in Figure 6c, as $d_{y}$ increases to 12 mm, the bifurcation point P_1_ continues to move to the left, and the bifurcation point P_2_ begins to move to the right. In addition, two saddle points S appear above and below P_1_, which divides $d_{x}$ into four intervals. Compared with Figure 6b, Figure 6c divides the $d_{x}>d_{xP1}$ interval into two $d_{xS}>d_{x}>d_{xP1}$ and $d_{x}>d_{xS}$ intervals. When $d_{x}>d_{xS}$, the system has only one stable equilibrium position and behaves as mono-stable. When $d_{xS}>d_{x}>d_{xP1}$, the system has five equilibrium positions, three of which are stable and two of which are unstable, and the system behaves as a tri-stable state. Figure 6d shows the static bifurcation diagram of the system when $d_{y}=16\;\mathrm{mm}$. The bifurcation points P_1_ and P_2_ disappear, and the two saddle points S move to the left and separate. When $d_{x}$ is on the right side of saddle point S, the system only has one stable equilibrium position, which is a mono-stable system. When $d_{x}$ is on the left side of point S, the system has three stable and two unstable equilibrium positions, and the state of the system is tri-stable.

In addition, it can be seen in Figure 6 that, given the magnet distance $d_{y}$, when the magnet distance $d_{x}<0.003\;m$, the bifurcation diagram only has stable solutions and no unstable solutions. This phenomenon illustrates that the system has only stable equilibrium positions and no unstable equilibrium positions when the distance between the end magnet of the combined beam and the external magnet is less than 0.003 m.

Then, the effect of $d_{x}$ on the static bifurcation characteristics of the system is analyzed in the ($d_{y}$, u) plane. When $d_{x}=10\;\mathrm{mm}$, the bifurcation point P_1_ and the saddle point S are divided $d_{y}$ into three segments, as shown in Figure 7a. When $d_{y}$ is taken to the right of the saddle point S, the system has only one stable equilibrium point and is in a mono-stable state. When the value of $d_{y}$ is on the left side of the bifurcation point P_1_, the system has two stable points and one unstable equilibrium point, presenting a bi-stable state. When $d_{y}$ is between the points P_1_ and S, the system has three stable and two unstable equilibrium points, at which time the system exhibits tri-stable characteristics. Figure 7b,c show the static bifurcation for $d_{x}=12\;\mathrm{mm}$ and 15 mm. As can be seen from the figure, with the gradual increase in $d_{x}$, the position of bifurcation point P_1_ is basically unchanged, while the two saddle points S gradually move to the left, resulting in a larger range of being in a mono-stable state and a smaller range of being in a tri-stable state of the system. When $d_{x}=20\;\mathrm{mm}$, it can be seen from Figure 7d that the two saddle points S of the system disappear, and the bifurcation point P_1_ gradually moves to the left. When $d_{y}$ is taken on the right side of P_1_, the system is a mono-stable system, and when $d_{y}$ is taken on the left side of P_1_, the system presents bi-stable characteristics.

Based on the above analysis, it can be concluded that the steady state of the system shifts between mono-stable, bi-stable, and tri-stable states as the magnet distances $d_{x}$ and $d_{y}$ vary in a given parameter plane. In the plane ($d_{x}$, u), with the gradual increase in $d_{y}$, the bi-stable region of the system becomes smaller until it disappears. In the plane ($d_{y}$, u), with the increase in $d_{x}$, the tri-stable region of the system gradually decreases until it disappears.

### 3.2. Analysis of System Potential Energy

The total potential energy of the system is the sum of magnetic potential energy and the strain potential energy of the system, which can be expressed as
(20)$$U=U_{M}+U_{R}$$

The magnet distances $d_{x}$ and $d_{y}$ play a decisive role in the magnetic potential energy, thus having a significant effect on the total potential energy of the system. From the above analysis of Figure 5, it is clear that the system presents a tri-stable state when $d_{x}$ and $d_{y}$ are in region C. Taking $d_{x}=13\;\mathrm{mm}$, $d_{y}=14$, 15.5, and 17 mm in region C, the potential energy curves of three different shapes of the tri-stable system are plotted as shown in Figure 8. The potential energy curves of the system all have three potential wells and are symmetrically distributed around u=0. The depths of the left and right potential wells are the same. The left and right potential wells are collectively called the side potential wells, and the middle potential well is called the central potential well. $\Delta U_{1}$ and $\Delta U_{2}$ denote the depths of the central and side potential wells, respectively. From Figure 8, it can be seen that, when $d_{y}=14\;\mathrm{mm}$, $\Delta U_{1}<\Delta U_{2}$; when $d_{y}=15.5\;\mathrm{mm}$, $\Delta U_{1}=\Delta U_{2}$; when $d_{y}=17\;\mathrm{mm}$, $\Delta U_{1}>\Delta U_{2}$.

### 3.3. Analysis of System Dynamics Response

#### 3.3.1. Influence of Starting Position on System Dynamic Response

According to the above analysis of potential energy, when the system exhibits the tri-stable state, its potential energy curve has three potential wells. According to the principle of minimum potential energy, when the system’s potential energy is a minimum value, the system is in a stable equilibrium state. Therefore, the lowest point of the potential well is the stable equilibrium position of the system. This paper mainly studies the system’s response characteristics when the initial velocity is zero, so the stable equilibrium position of the system is selected as the starting position. The effect of the starting vibration from different stable equilibrium positions on the dynamic response of the system is analyzed below. Taking the excitation acceleration $A=14\;m/s^{2}$, the excitation frequency F=9 Hz, the magnet distance $d_{x}=13\;\mathrm{mm}$, and the starting positions are the stable equilibrium position of the center and the right side, respectively. The Runge–Kutta method is used to numerically solve the dynamic equation of the system. In this paper, we mainly study the dynamic response characteristics of the system at a steady state, so the response time range is chosen as a period of 15–20 s after the system is stable.

When $d_{y}=17\;\mathrm{mm}$, $\Delta U_{1}>\Delta U_{2}$, and Figure 9 shows the phase portrait and time–voltage diagram of the system at different starting positions. When starting from the central equilibrium position, as shown in Figure 9a, the system cannot overcome the hindrance of the central potential barrier due to the large depth $\Delta U_{1}$ of the central potential well. Therefore, the system can only make small-amplitude movements within the central potential well, which results in a small response displacement and voltage. When starting from the right-side equilibrium position, as shown in Figure 9b, due to the small depth $\Delta U_{2}$ of the side potential well, the system can overcome the hindrance of the side potential barrier and enter the central potential well. Since the system has obtained greater kinetic energy at this time, the system can also overcome the large central potential barrier to achieve a large-amplitude movement between the wells, and the response displacement and voltage of the system are greatly improved.

When $d_{y}=14\;\mathrm{mm}$, $\Delta U_{1}<\Delta U_{2}$, Figure 10 shows the phase portrait and time–voltage diagram of the system when starting from different equilibrium positions. When the starting position is the central equilibrium position, as shown in Figure 10a, the system is located in the shallow central potential well, and the system can easily overcome the constraint of the central potential barrier and enter the side potential well. At this time, the system has obtained enough kinetic energy to pass over the larger side potential barrier. Thus, the system can perform a large-amplitude reciprocating motion between the three potential wells, and the response displacement and voltage of the system are relatively large. When the starting position is the right-side equilibrium position, as shown in Figure 10b, the system is located in the deep right-side potential well, and the system cannot escape the constraint of the right-side potential barrier. Therefore, the system can only perform small-amplitude movements in the right-side potential well, and the response displacement and voltage of the system are relatively small.

When $d_{y}=15.5\;\mathrm{mm}$, $\Delta U_{1}=\Delta U_{2}$, and Figure 11 shows the phase portrait and time–voltage diagram of the system. Since the central and side potential wells of the system have the same depth and are relatively small, the system can escape from the constraints of the potential barriers and make large-amplitude motions between the three potential wells, regardless of whether they start from the central or the right-side equilibrium position. At this moment, the displacement and voltage of the system both show larger response amplitudes.

From the above analysis, it can be concluded that the starting position of the system has a great influence on the system response characteristics. The suitable starting position can make it easier for the system to escape the constraints of the potential well, thus showing a high-energy output state and improving the energy harvesting efficiency of the system.

#### 3.3.2. Influence of Excitation Frequency on System Dynamic Response

The combined beam piezoelectric energy harvester designed in this paper is mainly applied to coal mine equipment. When coal mine equipment is running, it will generate vibration energy, and this vibration frequency is mainly concentrated in the low-frequency range. Therefore, it is necessary to analyze the influence of the excitation frequency on the system dynamic response.

Given that the starting position is the central stable equilibrium position, the magnet distance $d_{x}=13\;\mathrm{mm}$, $d_{y}=15.5\;\mathrm{mm}$, and the excitation acceleration $A=14\;m/s^{2}$, Figure 12 shows the excitation frequency–displacement bifurcation diagram of the system. According to the transition of the motion state of the system, the excitation frequency F can be divided into four intervals, which are 0–6.5 Hz, 6.5–11.5 Hz, 11.5–13.5 Hz, and 13.5–20 Hz. When F is below 6.5 Hz, the system is constrained to make a small-amplitude motion in the well, resulting in a small response displacement. When F is in the range of 6.5–11.5 Hz, the system achieves inter-well motion and is continuously in a large-amplitude response state, and the large-amplitude response bandwidth of the system is 5 Hz. When F is between 11.5 and 13.5 Hz, the system exhibits a state of chaotic motion, and the response displacement of the system is very unstable. When F is above 13.5 Hz, the system can only move in the well and is in a state of small-amplitude response. In addition, it can be found that the effective energy harvesting bandwidth of the system is significantly improved by introducing a nonlinear magnetic force.

It can also be found from Figure 12 that the response displacement of the system has an obvious jump when the excitation frequency F is 6.5 Hz, 11.5 Hz, and 13.5 Hz. With the increased excitation frequency F, the system’s attractor is transformed, resulting in this jump. When F=6 Hz, the attractor transforms from the intrawell periodic to the interwell periodic attractor. The system’s motion changes from the intrawell to the interwell motion, so the response displacement is greatly improved. When F=11.5 Hz, the interwell periodic attractor transforms into the chaotic attractor. The system changes from interwell to chaotic motion, so the response displacement is sometimes large and small. When F=13.5 Hz, the attractor transforms from the chaotic attractor to the intrawell periodic attractor. The system changes from chaotic to intrawell motion, significantly reducing the response displacement.

Figure 13 shows the phase portrait and time–voltage diagram of the system under different excitation frequencies. Figure 13a shows that, when F=3 Hz, the excitation frequency does not match the large-amplitude response frequency of the system, so the system can only perform a small-amplitude movement in the central potential well, and the output voltage of the system is only 5.8 V. As F increases to 9 Hz, as shown in Figure 13b, the excitation frequency at this time is within the large-amplitude response frequency range of the system, so the system realizes large inter-well motion, and the output voltage reaches 46 V. As F continues to increase to 12 Hz, as shown in Figure 13c, the excitation frequency is close to the large-amplitude response frequency of the system, so the system exhibits a state of chaotic motion, and the output voltage is sometimes large and small. As shown in Figure 13d, when F=18 Hz, the excitation frequency completely deviates from the large-amplitude response frequency of the system, so the system can only perform the small-amplitude intrawell motion, and the output voltage of the system is reduced to 8.6 V. It can be seen that the excitation frequency has a significant effect on the dynamic response of the system. The suitable excitation frequency makes it easier for the system to pass over the potential barriers and achieve large-amplitude periodic motions, thus improving the output performance of the system.

## 4. Experimental Verification

To validate the correctness of the simulation results, a test platform of the CTPEH system was built, and an experimental prototype was made in this paper. Figure 14 shows the experimental test platform. It is mainly composed of a computer, shaker (LT-50ST, ECON, Hangzhou, China), shaker controller (VT-9008, ECON, Hangzhou, China), power amplifier (VSA-L1000A, ECON, Hangzhou, China), laser vibrometer (LV-S01, SDPTOP, Shanghai, China), vibrometer controller (LV-S01, SDPTOP, Shanghai, China), vibration data collector (CoCo-80X, CRYSTAL, Silicon Valley, CA, USA), and oscilloscope (DSOX3024T, KEYSIGHT, Santa Rosa, CA, USA). Figure 15 shows the experimental prototype of CTPEH, which mainly consists of a base, fixture, PVDF piezoelectric film, combined beam, and magnet. One end of the combined beam is clamped and fixed to the base, while a magnet is attached to the other end. Two magnets are symmetrically fixed to the base, and the PVDF piezoelectric film is uniformly glued to the surface of the combined beam.

During the experiment, the experimental prototype is installed on the shaker, and the shaker controller generates a vibration signal through the control software on the computer. The power amplifier amplifies the signal and inputs it into the shaker, and the experimental prototype vibrates under the action of the shaker. The laser vibrometer and the vibration data collector are used to measure and collect the response velocity and displacement of the combined beam, respectively. The oscilloscope is used to collect the output voltage of the prototype.

Given the excitation frequency F=9 Hz, excitation acceleration $A=14\;m/s^{2}$, we must take the magnet distance $d_{x}=13\;\mathrm{mm}$, $d_{y}=17\;\mathrm{mm}$. Figure 16 shows the experimental phase portrait and time–voltage diagram under different starting positions. When the starting position is the central equilibrium position, as shown in Figure 16a, the system can only make small-amplitude periodic motions within the central potential well because it cannot cross the higher central potential barrier. The response displacement of the system is only 2.5 mm, and the output voltage is only 6.5 V. When starting from the right-side equilibrium position, as shown in Figure 16b. The system can easily pass over the lower right-side potential barrier and gain enough energy to cross the higher central potential barrier, thus achieving large-amplitude periodic motion in the tri-stable state. At this time, the response displacement of the system reaches 21 mm, and the output voltage is up to 52 V.

Given the excitation acceleration $A=14\;m/s^{2}$, magnet distance $d_{x}=13\;\mathrm{mm}$, and $d_{y}=15.5\;\mathrm{mm}$, the starting position is the central equilibrium position. Figure 17 shows the experimental phase portrait and the time–voltage diagram at different excitation frequencies. Figure 17a shows the system performing the tri-stable large-amplitude periodic motion at F=9 Hz, as the excitation frequency at this point matches the large-amplitude response frequency of the system. The response displacement of the system is 22 mm, and the output voltage is 55 V. Figure 17b shows that, at F=18 Hz, the system can only make the small-amplitude periodic motion in the well because the excitation frequency deviates from the large-amplitude response frequency of the system. The response displacement of the system is reduced to 4.5 mm, and the output voltage is lowered to 11.3 V.

It can be seen from the comparison that the results of the simulation analysis and experimental verification are relatively consistent, but there are some deviations in the numerical values. These deviations are mainly because of: (1) a deviation between the actual and simulated size of the combined beam and magnet; (2) the gravitational effect of the magnet which is not considered in the simulation, while it is real in the experiment; and (3) the curved beam part of the combined beam which will produce tensile deformation when moving, which leads to some deviations in the velocity and displacement results measured with the laser.

## 5. Conclusions

In this paper, the nonlinear magnetic and dynamical models of the CTPEH system are established. The effects of different magnet distances and external excitation conditions on the system’s static and dynamic response characteristics are analyzed by numerical simulations and experiments. Some of the main conclusions are summarized as follows:(1)The CTPEH system has the characteristic of a complex static solution bifurcation. Given the magnet distance $d_{y}$, there are four transitions in the steady state of the system as the magnet distance $d_{x}$ decreases. The first transition is a direct jump from mono-stable to bi-stable. The second is to jump from mono-stable to bi-stable and then to tri-stable. The third is to start from a mono-stable, first to a tri-stable, then jump to a bi-stable, and finally to a tri-stable. The fourth is a direct transition from mono-stable to tri-stable.(2)When the system exhibits a tri-stable state, the system has three potential energy wells. In the range of tri-stable, when the magnet distance $d_{x}$ is certain, adjusting the magnet distance $d_{y}$ can alter the depth of three potential wells of the system, thereby changing the shape of the potential energy curve of the system.(3)The different starting positions have a significant influence on the dynamic response of the system. When the starting position is near a shallow potential well, the system can easily overcome the potential barrier to achieve large-amplitude interwell motion, resulting in a large response displacement and output voltage. When the starting position is near a deep potential well, it is difficult for the system to overcome the constraints of the potential barrier. At this time, the system can only perform small-amplitude intrawell motion, which greatly reduces the response displacement and output voltage of the system.(4)The external excitation frequency has a great effect on the output characteristics of the system. As the excitation frequency increases, the system behaves as small-amplitude intrawell, large-amplitude interwell, chaotic, and small-amplitude intrawell motion states in that order. When the excitation frequency is within the large-amplitude response frequency range of the system, the system exhibits a high-energy output state, which is beneficial to the effective harvesting of energy.


## Figures and Tables

**Figure 1 micromachines-13-01465-f001:**
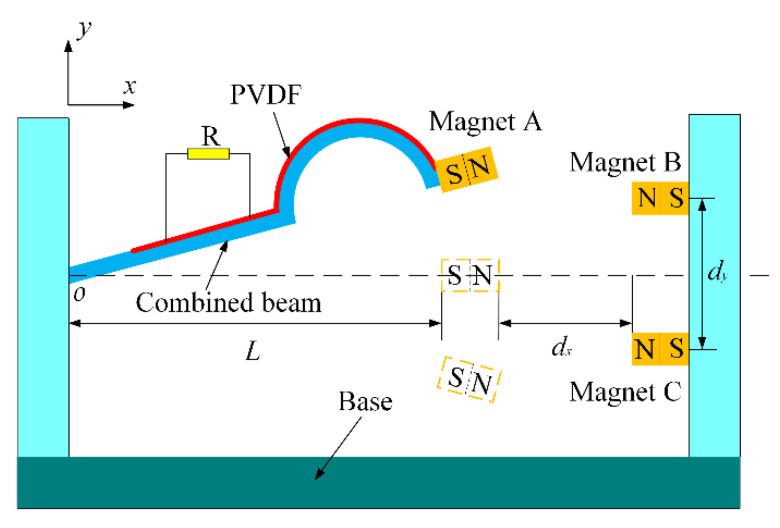
Schematic diagram of the structure of CTPEH.

**Figure 2 micromachines-13-01465-f002:**
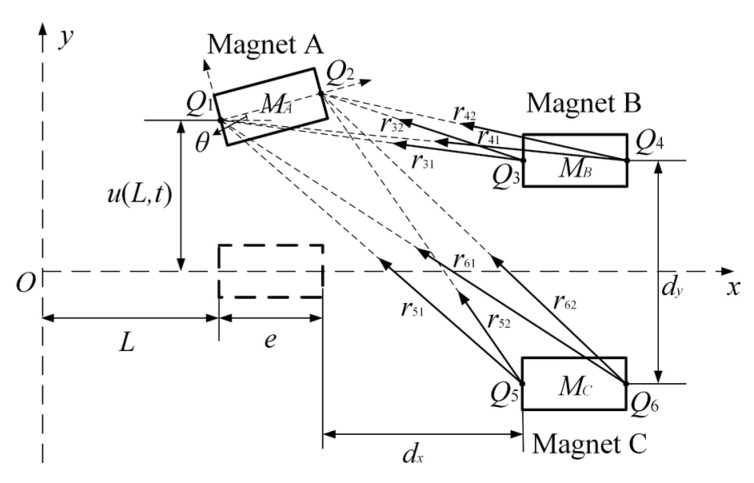
Schematic diagram of magnet position relationship.

**Figure 3 micromachines-13-01465-f003:**
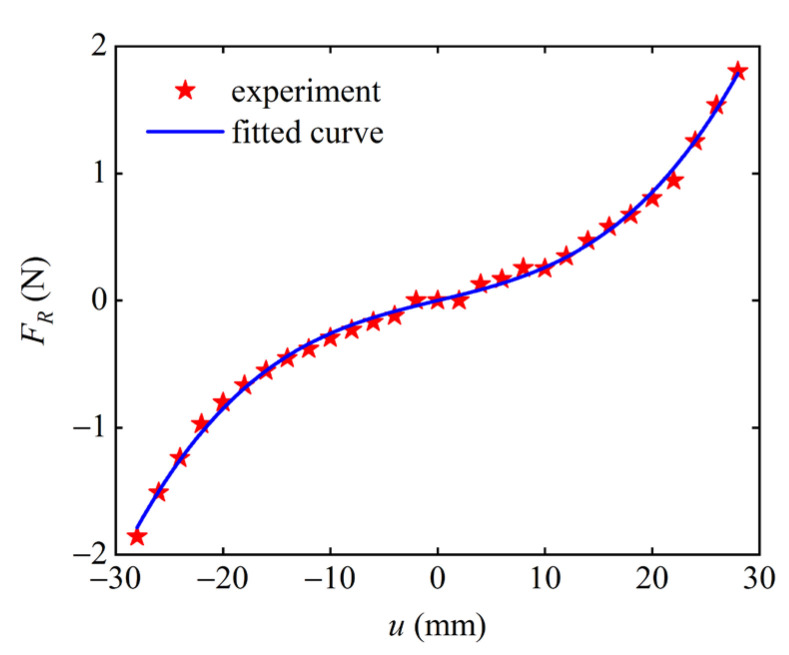
Nonlinear restoring force curve.

**Figure 4 micromachines-13-01465-f004:**
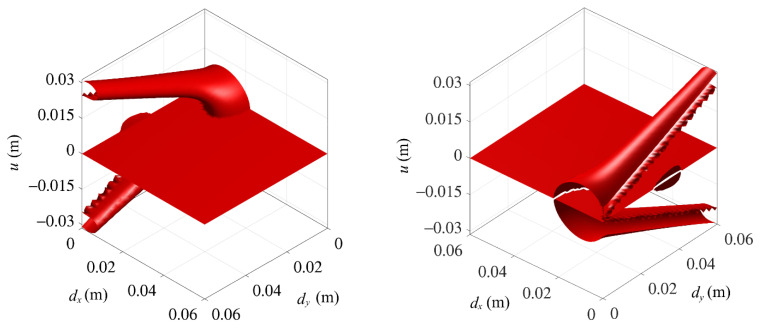
Schematic diagram of the static solution of the system.

**Figure 5 micromachines-13-01465-f005:**
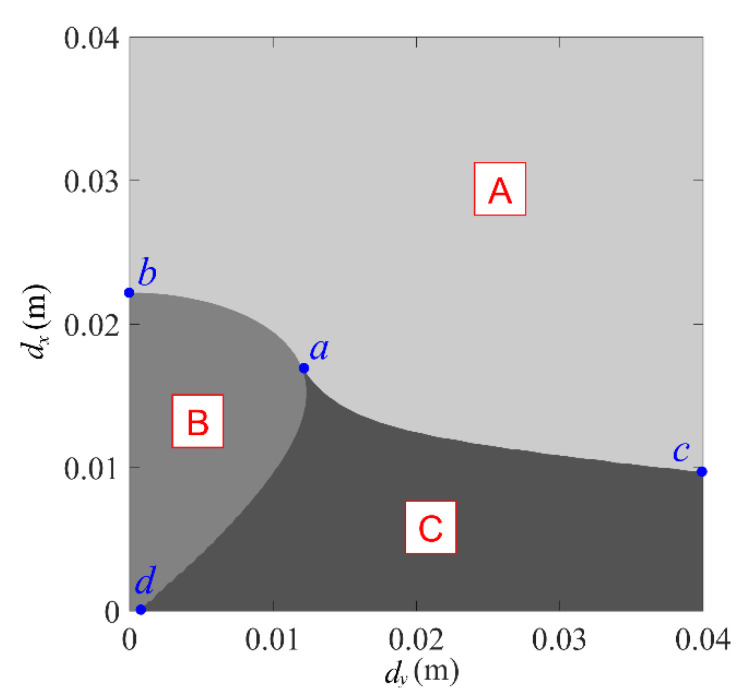
Bifurcated set of equilibrium positions.

**Figure 6 micromachines-13-01465-f006:**
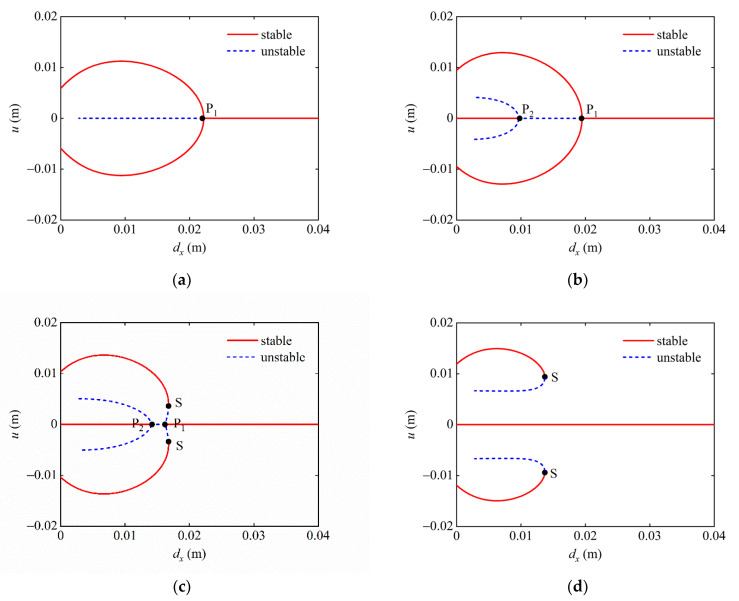
Static bifurcation diagram of the system under different magnet distances $d_{y}$: (**a**)$\;d_{y}=0\;\mathrm{mm}$; (**b**) $\;d_{y}=10\;\mathrm{mm}$; (**c**)$\;d_{y}=12\;\mathrm{mm}$; and (**d**) $\;d_{y}=16\;\mathrm{mm}$.

**Figure 7 micromachines-13-01465-f007:**
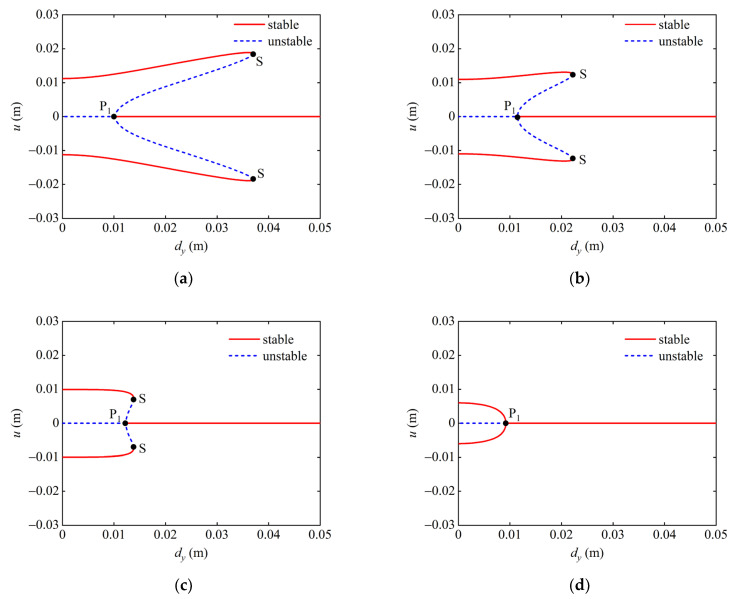
Static bifurcation diagram of the system under different magnet distances $d_{x}$: (**a**)$\;d_{x}=10\;\mathrm{mm}$ (**b**)$\;d_{x}=12\;\mathrm{mm}$; (**c**) $d_{x}=15\;\mathrm{mm}$; and (**d**) $\;d_{x}=20\;\mathrm{mm}$.

**Figure 8 micromachines-13-01465-f008:**
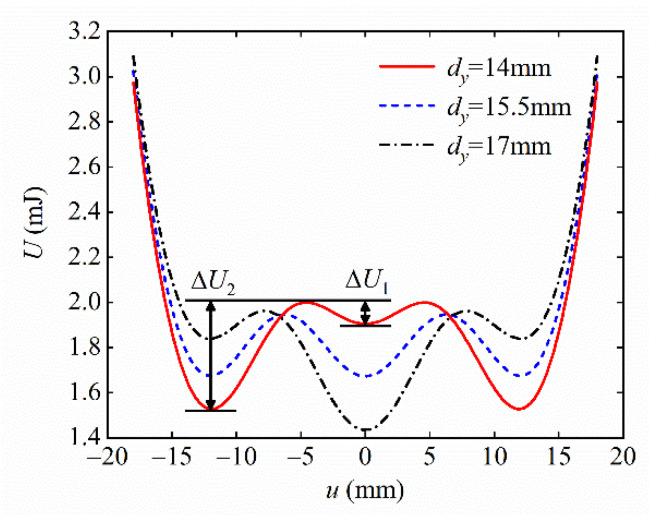
Potential energy curve of the system.

**Figure 9 micromachines-13-01465-f009:**
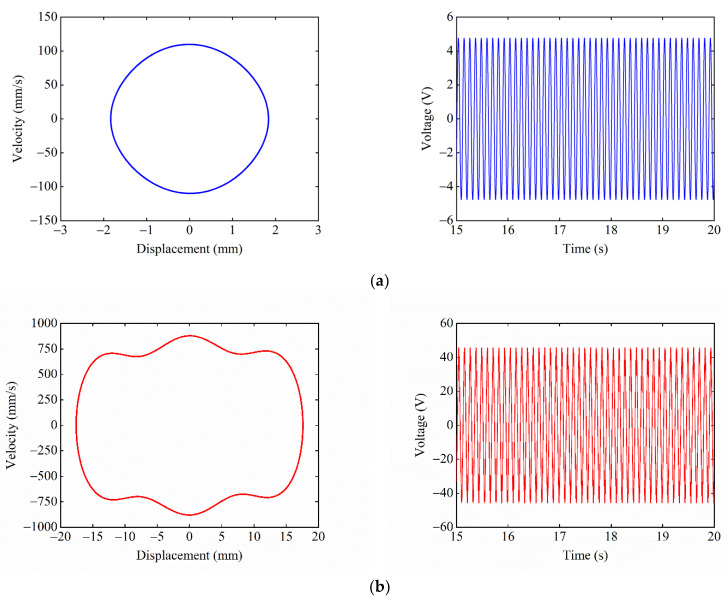
Phase portrait and time–voltage diagram of the system when $\Delta U_{1}>\Delta U_{2}$: (**a**) starting position is the central equilibrium position; and (**b**) starting position is the right-side equilibrium position.

**Figure 10 micromachines-13-01465-f010:**
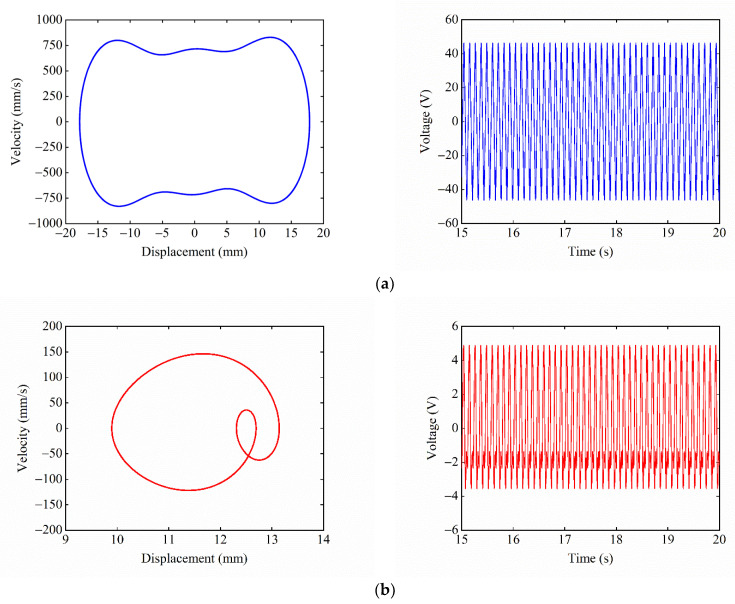
Phase portrait and time–voltage diagram of the system when $\Delta U_{1}<\Delta U_{2}$: (**a**) starting position is the central equilibrium position; and (**b**) starting position is the right-side equilibrium position.

**Figure 11 micromachines-13-01465-f011:**
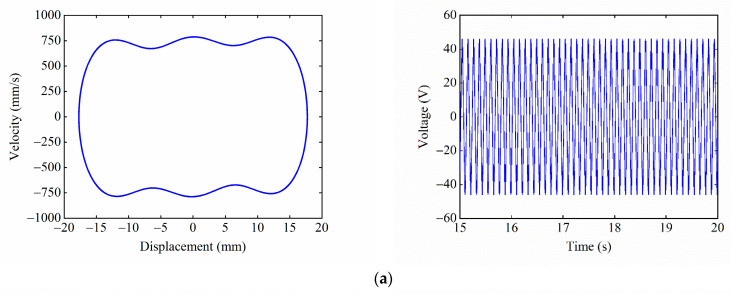
Phase portrait and time–voltage diagram of the system when $\Delta U_{1}=\Delta U_{2}$: (**a**) starting position is the central equilibrium position; and (**b**) starting position is the right-side equilibrium position.

**Figure 12 micromachines-13-01465-f012:**
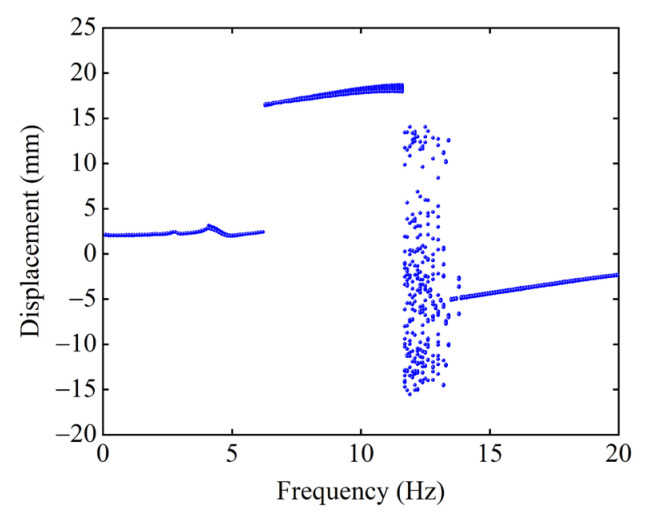
Excitation frequency–displacement bifurcation diagram of the system.

**Figure 13 micromachines-13-01465-f013:**
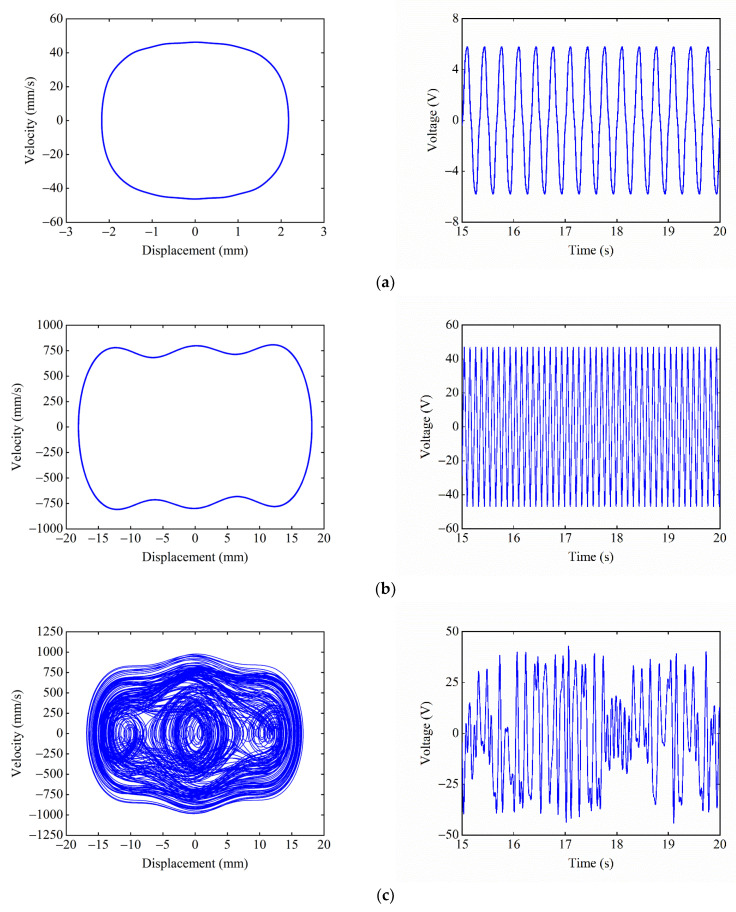
Phase portrait and time–voltage diagram of the system under different excitation frequencies: (**a**) F=3 Hz; (**b**) F=9 Hz; (**c**) F=12 Hz; and (**d**) F=18 Hz.

**Figure 14 micromachines-13-01465-f014:**
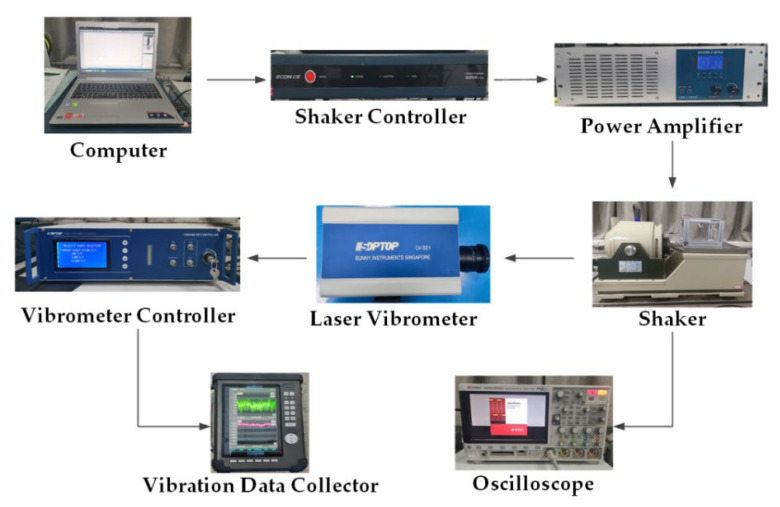
Experimental test platform.

**Figure 15 micromachines-13-01465-f015:**
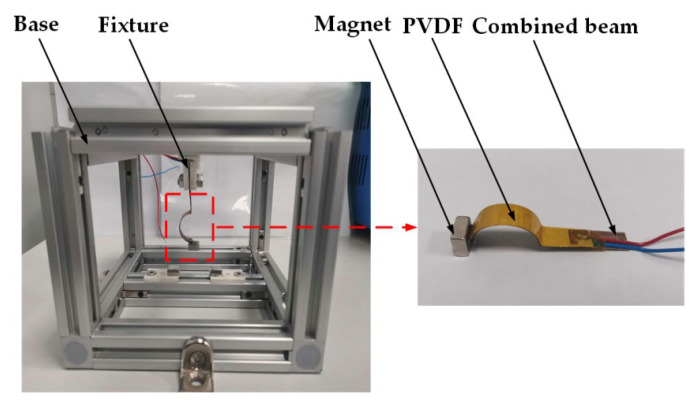
Experimental prototype of CTPEH.

**Figure 16 micromachines-13-01465-f016:**
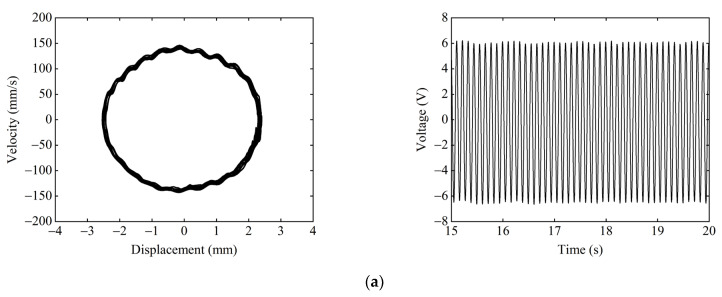
Experimental phase portrait and time–voltage diagram at different starting positions: (**a**) starting position is the central equilibrium position; and (**b**) starting position is the right-side equilibrium position.

**Figure 17 micromachines-13-01465-f017:**
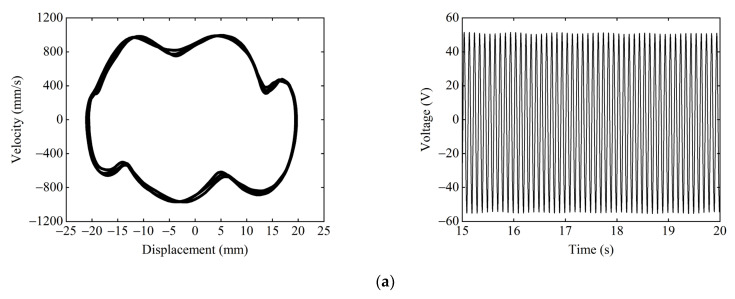
Experimental phase portrait and time–voltage diagram at different excitation frequencies: (**a**) F=9 Hz; and (**b**) F=18 Hz.

**Table 1 micromachines-13-01465-t001:** Structure and material parameters of CTPEH.

	Parameter	Value
	Length	40 mm
	Width	8 mm
Substrate layer	Height	0.2 mm
	Material density	8500 kg/m^3^
	Elastic modulus	128 × 10^9^ Pa
	Length	40 mm
	Width	8 mm
Piezoelectric layer	Height	0.11 mm
	Material density	1750 kg/m^3^
	Elastic modulus	3 × 10^9^ Pa
	Length	5 mm
	Width	10 mm
Magnet	Height	10 mm
	Material density	7500 kg/m^3^
	Magnetization	5.5 × 10^5^ A/m

## Data Availability

Not applicable.

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
