# Peer review of "Vibration Characteristics and Experimental Research of Combined Beam Tri-Stable Piezoelectric Energy Harvester"

_micromachines, 2022, doi:10.3390/mi13091465_

Round 1

Reviewer 1 Report

In the manuscript, a combined beam tri-stable piezoelectric energy harvester (CTPEH) was designed by introducing magnetic forces into the structure. TThe ordinary differential equations governing the nonlinear magnetic model were obtained for exploring the vibration characteristics of CTPEH. The effect of different magnet distances and excitation conditions on the static and dynamical bifurcation behaviors was discussed numerically and experimentally. In general, the work is well structured, and my overall assessment is positive. My main comments are presented as follows.

1. In section 2, for codimension-two bifurcation, the authors presented the results in dx-dy plane (see Figures 4 and 5). However, they did not provide how to classify the boundary of the bifurcation regions or the concerning proof.

2. In Figures 6 (b)-(d), there were some blanks between the vertical coordinate u and the dashing curves on the static bifurcation diagrams. Why? Please give the explanation in the content.

3. In Section 3.3.1, Lines 306-308 presented the comments related to potential wells and the initial positions. But why? Is there any literature to support it? How about the case that the initial velocity is not zero?

4. In Figure 11, the diagrams for time histories as well as phase maps seem repeated. If there is only one inter-well attractor, you may delete a pair. Otherwise, you may draw two inter-well attractors in the same plane by using different colors.

5. In Figure 12, from the the first interval to the second one, and from the second one to the third, there are obvious jump. The authors did not give the explanation. Thus it is hard for the readers to observe the route to chaos. Actually, it can be discussed combining the coexisting attractors induced by different initial displacement in Section 3.3.1.

6. Why using F to represent the frequency of the external excitation? It didn't appear in Equations (1)-(13). For a clearer illustration, please explain your ODE model in detail, and point out the frequency and amplitude of the external excitation in the model.

7. The second point of the conclusion (see Lines 464-469) was not clear or well-supported, as the potential energy curves were provided quantitatively for fixed values of dx and dy. Please provide qualitative conclusions.

8. As the manuscript discussed the influences of initial displacement on the dynamical behaviors, please added related references as well as their introduction.

9. The manuscript also needs some editing since it contains some grammatical errors.

Author Response

Dear reviewer, thank you very much for your helpful comments. We have revised the article based on your comments. The specific paper review responses are in the attachment below.

Reviewer 2 Report

This work proposes a combined beam tri-stable piezoelectric energy harvester (CTPEH) designed by introducing magnetic force into the combined beam structure. Overall, the article is well organized and the presentation is good. My concern is about the dynamic model of CTPEH at Section 2.2.3 (see Equation (7)). The authors consider the combined beam as a straight beam during the theoretical modeling. My suggestion is that the hypothesis of the model should be given before the theoretical modeling. An additional question, what is the merit of the curved beam compared with the straight beam in this design?

Author Response

(The authors gave the same response as above.)

Reviewer 3 Report

The authors presented a multi-stable energy harvester, coupling electromagnetic forces with vibration for improved energy harvesting. There are certain comments needed to be addressed. 

1. The introduction does not present the research's novelty, contribution, direction and position concerning the state-of-the-art. 

2. Figure 4 needs to be edited to a larger scale and better data representation.

3. Figure 5 needs better label colour.

4. Can you please comment on the power or power density level for this energy harvester?

Author Response

(The authors gave the same response as above.)

Round 2

Reviewer 1 Report

The authors have answered my questions properly. I have no new concerns.